# Thermosensation and Temperature Preference: From Molecules to Neuronal Circuits in *Drosophila*

**DOI:** 10.3390/cells12242792

**Published:** 2023-12-08

**Authors:** Meng-Hsuan Chiang, Yu-Chun Lin, Tony Wu, Chia-Lin Wu

**Affiliations:** 1Graduate Institute of Biomedical Sciences, College of Medicine, Chang Gung University, Taoyuan 33302, Taiwan; tony12185@gmail.com (M.-H.C.); lyc2001bs@gmail.com (Y.-C.L.); 2Department of Neurology, New Taipei Municipal TuCheng Hospital, Chang Gung Memorial Hospital, New Taipei City 23652, Taiwan; tonywu@cgmh.org.tw; 3Department of Biochemistry, College of Medicine, Chang Gung University, Taoyuan 33302, Taiwan; 4Brain Research Center, National Tsing Hua University, Hsinchu 30013, Taiwan

**Keywords:** *Drosophila melanogaster*, neuronal circuits, brain, thermosensation, temperature preference

## Abstract

Temperature has a significant effect on all physiological processes of animals. Suitable temperatures promote responsiveness, movement, metabolism, growth, and reproduction in animals, whereas extreme temperatures can cause injury or even death. Thus, thermosensation is important for survival in all animals. However, mechanisms regulating thermosensation remain unexplored, mostly because of the complexity of mammalian neural circuits. The fruit fly *Drosophila melanogaster* achieves a desirable body temperature through ambient temperature fluctuations, sunlight exposure, and behavioral strategies. The availability of extensive genetic tools and resources for studying *Drosophila* have enabled scientists to unravel the mechanisms underlying their temperature preference. Over the past 20 years, *Drosophila* has become an ideal model for studying temperature-related genes and circuits. This review provides a comprehensive overview of our current understanding of thermosensation and temperature preference in *Drosophila*. It encompasses various aspects, such as the mechanisms by which flies sense temperature, the effects of internal and external factors on temperature preference, and the adaptive strategies employed by flies in extreme-temperature environments. Understanding the regulating mechanisms of thermosensation and temperature preference in *Drosophila* can provide fundamental insights into the underlying molecular and neural mechanisms that control body temperature and temperature-related behavioral changes in other animals.

## 1. Introduction

The physiological processes of all organisms are significantly affected by temperature. The ability to sense, respond, and adapt to external environmental threats is the most critical factor that has evolved to protect the existence of organisms [1]. Suitable temperatures can affect animal responsiveness, movement, metabolism, growth, and reproduction, whereas extreme temperatures can be injurious and potentially lethal. Thermosensation is important not only for endotherms but also for ectotherms. Thermosensation is one of the most ancient sensory processes in all animals. The thermal information is generated by thermoreceptors and transported to animal’s brain. The translation of thermal energy into electrical signal is mediated by these thermoreceptors that are sensitive to a specific range of temperatures [2]. Understanding the mechanisms involved in regulating body temperature (*Tb*) in different environments is critical for survival. Physiological factors, including morphology, heat adaptation, sex, and age, affect *Tb* regulation [3]. Brown adipose tissue (BAT), shivering, and the constriction of blood vessels play important roles in the defense against cold temperatures. Conversely, during hot temperatures, thermogenesis decreases, sweat production increases, and blood vessels dilate [4]. Animal behavior also plays a crucial role in *Tb* control. The most basic thermoregulatory behavior involves seeking cold or warmth. Animals move between microenvironments in their habitats to alter the rate of heat loss or absorption. Besides basic thermoregulatory behaviors, more complex strategies, such as making nests or burrows, have been employed to establish a suitable temperature environment [4].

In mammals, environmental temperature is sensed by primary somatosensory neurons in the skin. These neurons subsequently relay this information to the dorsal horn (DH) of the spinal cord. The DH contains neurons projecting to the lateral parabrachial nucleus (LPB) of the brainstem. Finally, third-order neurons located within the LPB project to the preoptic area (POA) of the hypothalamus (Figure 1A,B). In addition, the activation of LPB neurons can be induced by changes in skin temperature. The external lateral subdivision of the LPB (LPBel) receives cold-temperature information from the DH and transmits it to the median preoptic area (MnPO). LPBel neurons are necessary for cold-defense responses, such as shivering and BAT thermogenesis. In contrast, the dorsal subdivision of the LPB (LPBd) receives hot-temperature information from the DH and predominantly transmits it to the MnPO. LPBd neurons are necessary for hot-defense responses, such as cutaneous vasoconstriction and inhibition of BAT thermogenesis [3,4,5].

*Drosophila melanogaster* can achieve a desirable *Tb* through ambient temperature fluctuations, sunlight exposure, and behavioral strategies [6,7,8,9,10]. The availability of powerful tools for genetic manipulations has allowed scientists to understand detailed molecular mechanisms and neuronal circuits that regulate thermosensation and temperature-preference behaviors in *Drosophila*. Since the prominent signaling pathways that regulate neuronal functions in mammals are highly conserved in *Drosophila*, scientists have been using this model organism to study thermosensation. Temperature-related behaviors in *Drosophila* include temperature preference rhythm (TPR), the temperature preference of aged flies, and feeding-state-dependent temperature preference. This review discusses current findings regarding the genes, circuits, and molecular mechanisms of thermosensation and temperature-related behaviors in *Drosophila*.

## 2. Thermosensation in *Drosophila*

### 2.1. The Molecular Basis of Thermosensation

Most ion channels belong to the transient receptor potential (TRP) family. TRP is conserved across species, from humans to fruit flies. In mammals, these channels can be classified into more than fifty subtypes, which are further divided into seven subfamilies based on the homology of their amino acid sequences. These subfamilies include vanilloid (TRPV1-6), canonical (TRPC1-7), melastatin (TRPM1-8), non-mechanoreceptor potential C (NOMP-like, TRPN1), long TRP ankyrin (TRPA1), polycystins (TRPP1-5), and mucolipins (TRPML1-3) [11,12,13,14,15,16,17]. TRP channels exhibit diverse modes of activation through temperature, mechanical, and chemical stimuli, thereby facilitating the modulation of cellular functions via an inward cation current [18,19]. Consequently, the TRP family regulates a wide range of physiological processes, such as responses to environmental stimuli and functions related to vision, hearing, taste, and thermosensation [20]. In mammals, TRPV1-4 receptors are responsible for receiving heat stimuli, whereas TRPA1 and TRPM8 receptors are specialized in detecting cold temperatures. While TRPV1-4 receptors operate within the innocuous temperature range, it is noteworthy that TRPA1, TRPV1, and TRPV2 are capable of functioning within the noxious temperature range [21,22,23,24,25,26,27,28,29,30,31,32,33,34,35,36,37,38]. Furthermore, a previous study showed that TRPM3 receptors serve as sensors for detecting nociceptive heat while TRPM2 receptors serve as warmth sensors in mice [39,40,41].

Manning et al. first demonstrated the existence of TRP channels in *Drosophila* while investigating a spontaneous mutation that exhibited a transient response to prolonged intense light. This mutation was initially named “type A” [42]. Subsequently, it was found that this mutant, referred to as “transient receptor potential” or TRP is defective for light transmission [43]. TRP channels play a role in temperature sensing in *Drosophila*. In addition to thirteen TRP channels, *Drosophila* also possesses four *TRPA* homologs: *dTrpA1*, *painless*, *pyrexia*, and *water witch* [44]. In terms of amino acid identity, *dTrpA1* is 32% identical to and 54% comparable with its ortholog in mammals [45]. The dTRPA1 channel is expressed in several classes of peripheral sensory neurons and numerous groups of central neurons [46,47,48]. *painless* is expressed in the larval peripheral nervous system [49] and multiple regions of the adult brain, such as the mushroom body (MB), ellipsoid body (EB), projection neurons (PNs) of olfaction, and the pars intercerebrails (PI) [50,51,52,53]. *pyrexia* is ubiquitously expressed along the dendrites of a subset of peripheral nervous system neurons [54]. *water witch* is expressed in neurons associated with specialized sensory hairs located in the third segment of the antenna [55,56]. These neurons extend to the regions of the central nervous system that are involved in the perception of mechanical stimuli [57,58]. The functionality of *water witch* is essential for the detection of humid air [57]. In contrast to mammalian *TrpA1*, *dTrpA1*, *painless*, and *pyrexia* have no role in regulating cold avoidance. However, *dTrpA1* is also recognized as a receptor for detecting warmth [46,59,60,61]. Additionally, temperatures over 38 and 42 °C can be detected by *painless* and *pyrexia*, respectively [49,54,61,62,63].

Gallio et al. identified *brv1*, *brv2*, and *brv3* as potential receptors for cold stimuli [38]. The *brivido* genes encode channels homologous to the mammalian TRP channels [38,64]. These channels are specifically found in neurons that are sensitive to cold temperatures, capable of detecting drops in temperature up to 11 °C. These channels are not expressed in neurons that respond to heat and can detect temperatures exceeding 25 °C. Notably, these genes are co-expressed in the same sensory neurons, suggesting a potential collaborative role in enhancing the perception of cold stimuli. The elimination of *brv1* or *brv2* results in a diminished neuronal response to temperature decline in calcium imaging experiments, indicating their essential role in cold sensing. Furthermore, *brv1/2/3* genes are also crucial for cold avoidance in the two-choice behavior assay, where flies are faced with the decision of selecting between 25 °C and a lower temperature [38]. The two-choice behavior assay is used to identify temperature avoidance or temperature preference in *Drosophila* and involves a two-choice thermoelectric device. In this assay, the avoidance index is calculated by comparing the fly behavior at an experimental temperature to that at a constant temperature (25 °C) (Figure 2A–C).

It has been shown that the presence of additional receptor proteins besides TRP channels is responsible for temperature sensitivity. In *Drosophila* larvae, the role of the photoreceptor rhodopsin, a canonical G protein-coupled receptor in thermosensory neurons, remains unclear. However, recent studies have indicated that it might play a role in temperature responses [67]. In adult *Drosophila*, the gustatory receptor, GR28B(D), plays a role in detecting high temperatures [68]. Invertebrates possess seven transmembrane proteins known as gustatory receptors [69]. Gustatory receptors, along with olfactory receptors, constitute a distinct gene family and have been extensively studied because of their involvement in the chemoreception of sweet and bitter taste, food odors, carbon dioxide, and various other substances [69,70,71]. A previous study showed that *GR28B(D)* is expressed in hot cells, the neurons responsible for thermosensation. Furthermore, when *GR28B(D)* is ectopically expressed in different cell types, it confers the ability to sense temperature, particularly warmth. These findings strongly indicate that *GR28B(D)* functions as a sensor for detecting warmth [68]. Although *GR28B(D)* and *dTrpA1(B)* mediate different forms of thermotactic behaviors and have different expression patterns, it can be postulated that these receptors function as sensors for high temperatures. Although *dTrpA1* is expressed in anterior cell (AC) neurons of the anterior brain, *GR28B(D)* is expressed in peripheral hot cells located at the base of the aristae. *GR28B(D)* is necessary for rapid negative thermotaxis, whereas *dTrpA1* is necessary for slow thermotaxis on a shallow and broad thermal gradient [46,68,72,73].

The ionotropic receptors (IR) IR21a and IR25a, members of the ionotropic glutamate receptor (iGluR) family, have also been implicated in thermosensation [74,75]. Invertebrate IRs commonly act as the receptors for various acids and amines. Although the specific chemoreceptor function of IR21a has not been identified yet, it is conserved in insects, indicating its potential involvement in other sensory modalities [76,77]. The most conserved insect IR across all species is IR25a. In *Drosophila*, IR25a is expressed in various classes of chemosensory neurons with distinct chemical characteristics. Additionally, IR25a acts as a co-receptor that combines with other stimulus-specific IRs to generate chemoreceptors with different specificities [74,78,79]. It has been shown that both IR21a and IR25a are required for mediating larval dorsal organ ganglion (DOG) responses to cold and regulating cold-avoidance behavior. By ectopically expressing IR21a, it has been shown that IR25a-dependent cold sensitivity is involved, indicating that these two receptors collaborate for cold perception [75] (Table 1).

### 2.2. The Neuronal Basis of Thermosensation

*Drosophila* senses temperatures through numerous temperature-sensing neurons. During the larval stage, the anterior tip of the larval head contains neurons that are specifically responsible for temperature detection. The cell bodies of these temperature-sensing neurons are located in two ganglia: the DOG and the terminal organ ganglion (TOG) [70,75,80,81,82,83]. Dendritic projections of these neurons extend towards two anatomical structures located beneath the larval body wall, known as terminal organs and dorsal organs, which have the potential to gather external temperature information [84]. Particularly, the responses of DOG and TOG to cold temperatures (below 25 °C) have been observed. DOG expresses ionotropic cold receptors IR21a and IR25a, which exhibit decreased activity as the temperature rises [75]. It has been found that larval cold-avoidance behavior ceases when the temperature-sensing DOG neurons are deactivated. However, the inhibition of TOG did not yield the same results. Both DOG and TOG project to the antennal lobes and establish connections with secondary neurons, resembling the olfactory neurons present in the antennal lobes of both adults and larvae [85]. Furthermore, neurons present in the lateral body wall also show sensitivity to temperature within the range of 10–40 °C. Notably, most of these neurons are stimulated by increasing temperatures and their activity is suppressed by decreasing temperatures [82].

In adult *Drosophila*, both peripheral and internal neurons are present and can detect temperature changes. Peripheral neurons, specifically primary sensory neurons, are responsible for sensing increases (hot) or decreases (cold) in temperatures. Among the primary sensory neurons located in the antennae, three are sensitive to hot temperatures, and three are sensitive to cold temperatures. Additionally, at the base of each arista, three hot-sensing neurons express the gustatory receptor, GR28B(D) [68]. Furthermore, in the sacculus of the third antennal segment and at the base of the aristae, three cold-sensing neurons express *brv1* [38] (Figure 1C). However, a more recent study indicated that the presence of *brv1* may not be necessary for temperature sensitivity in aristal cool cells [86]. Instead, thermosensation in cool cells is mediated by IR21a, IR25a, and IR93a [86]. Collectively, these six neurons function together to perceive temperatures. There is an anti-inhibition effect between hot-sensing and cold-sensing neurons. Hot-sensing neurons are suppressed by cold stimuli, whereas cold-sensing neurons are suppressed by hot stimuli. Consequently, the removal of hot or cold stimuli can have opposing effects, resulting in the perception of cold or hot sensations [38,87]. These primary sensory neurons that detect temperature project to the proximal antennal protocerebrum (PAP), where they form synapses with downstream PNs to facilitate further processing. It is important to highlight that the targets of these primary sensory neurons differ within peripheral afferent pathways. This observation suggests that the processing of hot and cold information occurs independently [38] (Figure 1C). Besides peripheral neurons located on the surface, additional internal neurons can also detect temperature. Within the anterior region of the brain, AC neurons are sensitive to warmth and express *dTrpA1*. These dTRPA1-expressing neurons extend their projections to the PAP, similar to the peripheral primary sensory neurons mentioned previously [46].

A specific group of secondary neurons, known as thermosensory PNs, in the brain is responsible for receiving signals from both hot and cold thermoreceptors [88]. These PNs have dendrites located in the proximal antennal protocerebrum and their axons extend to higher brain regions. One type of PN, known as warm-PNs, is activated by an increase in temperature and suppressed by a decrease in temperature. Warm-PNs exhibit a stronger response to rapid increases in temperatures than to slow increases in temperatures and have a capacity for adaptation to sustained temperature increases. Two distinct types of PNs, fast-cool and slow-cool, have been reported in response to cooling and are inhibited by warming stimuli. Fast-cool PNs display a significant adaptation to sustained temperature decreases and demonstrate higher responsiveness to rapid cooling than to slow cooling. In contrast, slow-cool-PNs exhibit little adaptation to prolonged temperature decreases and respond similarly to both slow and fast cooling. Fast-cool PNs project to the lateral protocerebrum, whereas slow-cool PNs project to the MB. Furthermore, the two types of cool PNs encode different cool inputs. Fast-cool PNs receive inputs from cool cells in the arista, whereas slow-cool PNs primarily receive inputs from cool cells in the sacculus, with weaker inputs from cool cells in the arista. Additionally, two PNs, known as called warm-cool PNs, extend to the lateral protocerebrum. These neurons exhibit a rapid response to changes in temperature, particularly fast changes, whereas their response to sustained stimuli is strongly adaptive [88].

A specific subset of dopaminergic neurons (DANs) upstream of the MB in the brain exhibits responsiveness to cooling stimuli. The MB is a large brain structure comprising approximately 2000 MB neurons (MBn), known as Kenyon cells, in each brain hemisphere. MBn can be further classified into γ, αβ, and α′β′ MBn based on the distribution of their axons [89]. There are two clusters of DANs: the protocerebral anterior media (PAM), whose axons project to the MBn horizontal lobes, and the protocerebral posterior lateral 1 (PPL1), whose axons project to the MBn vertical lobes [90,91,92]. These DANs regulate behavior by modulating MB neuronal activity through four types of dopamine receptors: Dop1R1 (also known as dD1 or DUMB), Dop1R2 (also known as DAMB), Dop2R (also known as DD2R), and DopEcR (noncanonical DA/ecdysteroid receptor) [90,91,92]. There are 21 types of MB output neurons (MBONs) that elaborate on the segregated dendritic arbors along the parallel axons of Kenyon cells. The MBn lobes are subdivided into 15 distinct compartments containing the segregated dendrite arbors of MBONs [91]. PPL1 neurons display primary phasic responses to cooling, with a preference for the initial decrease in temperature [93]. While there is currently no direct evidence supporting the notion that these neurons receive inputs from the previously described thermal sensory neurons or thermal PNs, experiments involving the removal of antennae have demonstrated a reduction in the response of PPL1 neurons in the γ2α′1 region of the MB when temperature is decreased. Additionally, the removal of both antennae and maxillary palps has been found to diminish the responses of DANs in both the γ1pedc and γ2α′1 regions of the MB lobe. These findings suggest that cold-responsive DANs may receive inputs from thermal sensory neurons located in the antennae and that thermal sensors also exist in the maxillary palps [93] (Table 2).

## 3. Temperature-Related Behavior in *Drosophila*

*Drosophila* is an ectotherm, and its preferred temperature (*Tp*) closely matches its selected environmental temperature [94]. Behavioral strategies for thermoregulation play a pivotal role in maintaining metabolism, energy storage, survival, temperature homeostasis, and responses to extreme environments [95,96]. *Tp* is not constant for adult flies or larvae. They can vary according to circadian rhythms, aging, and feeding status [56,66,94,97,98,99]. Furthermore, the acute avoidance of extreme temperatures is crucial for preventing or minimizing damage to flies. In this context, we will discuss temperature preference behavior under different physiological conditions and thermal nociception behavior.

### 3.1. Temperature Preference in Aged Drosophila

Aging is a universal biological phenomenon experienced by all living organisms. Several animal models have demonstrated a decline in physiological functions with age, including reproductive functions, muscle strength, cognitive impairment, and age-related diseases [100]. Several studies have used animal models such as rodents, *Drosophila*, and *Caenorhabditis elegans* to characterize physiological changes during aging [101]. In rats, age-dependent changes in thermoregulation have been linked to a decrease in the number of dopaminergic neurons. Specifically, the brain’s levels of the key enzyme tyrosine hydroxylase (TH), which plays a crucial role in dopamine synthesis, significantly decreases in rats during aging. This decline in TH levels leads to a decreased resistance to cold stress [102,103].

The concept of a temperature gradient device was developed in 1922 [84]. Researchers have modified the temperature gradient devices according to their experiments or to overcome mechanical obstacles. Sokabe et al. developed a continuous linear gradient by placing black aluminum plates on aluminum blocks, and each plate was set to a distinct temperature using circulating water from a bath [104]. The test plate was divided into 2 cm wide zones ranging from 18 to 28 °C with a gradient of 2 °C. During the experiment, 10–20 min after releasing approximately 150 larvae flies between the 22 and 24 °C zone, the distribution of larvae was calculated as follows: (number of larvae in a given 2 cm zone)/(total number of larvae in six zones) × 100%. However, the late third-instar larvae began to climb on the surface to escape from the food and prepare for pupation. To avoid the juxtaposition of this zone with the edge of the plate and test whether the selection of the 18 °C zone was strong, the authors created a temperature gradient in which the 18 °C zone was in the center and the warmer zones radiated out symmetrically on both sides [104]. Using the temperature gradient assays mentioned above, Sokabe et al. reported that the second-instar and early third-instar larvae (48 and 72 h after egg laying) prefer 24 °C, but this preference shifts to stronger biases for 18–19 °C in mid- and late-third-instar larvae (96 and 120 h after egg laying) [104]. The authors also showed that the rodopsin1 (*rh1*) null mutant (*ninaE^I17^*) larvae were impaired in the 18 °C selection at the mid-third-instar stage, but not in the late-third-instar stage [104]. This is consistent with a study by Shen et al. which showed that third-instar larvae prefer to stay at 18 °C through the thermosensory signaling pathway, which is light-dependent and depends on the guanine nucleotide-binding protein (G protein, Gq), phospholipase C (PLC), and the transient receptor potential TRPA1 channel [67]. Through behavioral assays of rhodopsin (*rh*) gene mutant screening, Sokabe et al. showed that both *rh5* and *rh6* are required for *Tp* in late-third-instar larvae [104]. Furthermore, *rh5* and *rh6* expression in *dTrpA1*-positive neurons in the brain and periphery induces the Gq/PLC/TRPA1 signaling cascade for thermotaxis in a light-independent manner in late-third-instar larvae [104]. However, whether rhodopsin acts as a direct thermosensor in *Drosophila* remains unclear.

Tyrrell et al. identified additional molecular mechanisms underlying the *Tp* [105]. Unlike the devices mentioned above, the authors utilized another larval temperature gradient assay for their study. In this assay, the authors placed an aluminum plate on the top of an ice bath on the right side and a hot plate on the left side set to approximately 70 °C. An agar gel was placed in the middle of the aluminum plate. This setup generated a continuous linear gradient from 13 to 31 °C, divided into 1 cm wide zones with 1 °C intervals. During the experiment, 10–15 min after releasing 20–35 larvae in the 22 °C zone, the larval distribution was calculated as follows: (number of larvae in the temperature zone)/(total number of larvae) × 100% [105]. Using a larval temperature gradient assay, the authors found that larvae sought a lower temperature during the late-third-instar stage, which is consistent with the study by Sokabe et al. Next, by silencing the dorsal organ cool cells (DOCCs) using a synaptic neurotransmitter blocker tetanus toxin light chain (TNT), the authors found that DOCCs are critical for cold avoidance in the early third-instar, but not in the late-third-instar stage [105]. Calcium imaging data further indicated that DOCCs exhibit reduced cooling responses in the late-third-instar stage [105]. These results suggest that the cool responses of DOCCs are significantly reduced during the late-third-instar stage, which enabled the larvae to remain at a lower temperature. Finally, through immunostaining and behavioral assays, the authors found that in early third-instar larvae, IR21a, IR25a, and IR93a are highly expressed in DOCCs, which led to an increase in cold avoidance [105]. In contrast, in late-third-instar larvae, IR21a, IR25a, and IR93a showed lower expressions in DOCCs and consequently, a reduced cold avoidance [105].

In contrast to larvae, few studies have addressed the age-dependent changes in adult flies. A study by Shih et al. revealed that adult fruit flies prefer to stay at lower temperatures and the MB is involved in regulating age-dependent temperature preference changes [66]. In this study, a temperature gradient device was used for the behavioral assay (Figure 2D–F). Shih et al. identified a microcircuit, the MB β and β′ system, that regulates *Tp* as flies age [66] (Figure 3A). First, using a behavioral assay, the authors found that flies preferred lower temperatures during aging. They also found that TH-immunopositive signals in the tip of the β′ lobe were strongest among all MB regions in young flies (7 days after eclosion) but significantly decreased in aged flies (21 days after eclosion) [66]. Live brain calcium imaging also demonstrated that the β′ lobe and β lobe strongly responded to cold stimuli [66]. Next, through silencing the neural activity, the authors found that the β′ system, including PAM-β′2, α′β′ MBn, and MBON-β′2mp, and the β system, including PAM-β1/β2, αβ MBn, and MBON-β2β′2a, are all involved in cold avoidance in young flies. In aged flies, PAM-β′2, α′β′ MBn, and MBON-β′2mp exhibited a decreased response to cold stimuli and the activity of these neurons is not required for cold avoidance [66]. These results suggest that both β and β′ systems are required in young flies to establish normal cold avoidance and set the *Tp*. However, in aged flies, TH expression is decreased in PAM-β′2 which leads to a significant reduction in the dopaminergic modulation in the β′ system. Therefore, the β system takes over the regulation to play a major role in cold avoidance in aged flies [66]. Several studies have suggested that when flies are exposed to low-temperature environments, their metabolic rates decrease, potentially increasing their longevity [106,107]. This may explain why older flies prefer lower temperatures.

### 3.2. Temperature Preference Rhythm

Body temperature rhythm (BTR) is a phenomenon in which the body temperature fluctuates over 24 h. BTR is highly relevant in maintaining homeostasis by regulating metabolic energy and sleep [108,109,110] (Figure 3A). At least 300 studies have addressed BTR in various species [110]. In humans, the core body temperature (CBT) and distal skin temperature (DST) serve as marker rhythms [111,112,113,114]. Both are subjected to numerous environmental and physiological influences that mask their rhythms, including physical activity, body position, light exposure, environmental temperature, and sleep [115,116,117,118,119,120,121,122,123]. CBT is approximately 37 °C in healthy humans, with nearly 1 °C sinusoidal circadian fluctuations. In mammals, the body temperature is controlled by a biological clock. The suprachiasmatic nucleus (SCN) in the brain controls all biological rhythms via the BMAL1:CLOCK heterodimer loop, which regulates clock-controlled genes [124,125,126,127,128].

Both mammals and ectotherms generate BTR through behavioral strategies to regulate body temperature [110,129]. *Drosophila* exhibits a robust temperature preference rhythm (TPR) through which flies select a suitable environmental temperature to set their body temperature [130,131]. The TPR increases during the day, peaks in the evening, and decreases at night [130,131], suggesting that *Drosophila* TPR is similar to the BTR [130,131]. *Drosophila* TRP is divided into four time zones in a day: daytime zeitgeber time (ZT1-ZT12), night-onset (ZT9-ZT12, ZT13-ZT15), dark time (ZT12-ZT24), and predawn (ZT22-ZT24) [131].

In molecular terms, two basic helix-loop-helix (bHLH) regulator proteins, the clock (*Clk*) and cycle (*cyc*), act as rhythm-driving factors by directly activating the transcription of many genes, including period (*per*) and timeless (*tim)* [132]. PER and TIM contribute to a negative feedback loop in the molecular cycle, thereby regulating circadian rhythms. *per* and *tim* mRNA levels peak in the evening, while protein levels reach their highest levels around the daybreak. PER and TIM undergo several post-translational modifications that critically influence their stability and ability to accumulate and affect their function. Over individual time courses, PER and TIM enter the nucleus and terminate the daily actions of CLK and CYC during transcriptional activation. The subsequent degradation of TIM and PER allows for the resumption of CLK-CYC-mediated transcriptional activation, starting another daily cycle [132,133,134].

At the neuronal circuit level, clock neurons in the *Drosophila* brain are divided into two groups: the lateral neurons (LN) and the dorsal neurons (DNs) [135]. Traditionally, LNs are considered key for controlling daily patterns of rest and activity [132,135,136,137]. LN can be further divided into three groups: large ventrolateral neurons (lLNv), small ventrolateral neurons (sLNv), and dorsolateral neurons (LNd) [135]. In contrast, the DN is thought to play subtle roles in modulating circadian rhythmicity [132,138]. Similar to LN, DN can be divided into three recognized groups: DN1, DN2, and DN3 [132]. These two groups of clock neurons control physiological functions and behavior; however, the interaction between these groups remains largely unknown. Additionally, a class II G-protein-coupled receptor, the pigment-dispersing factor receptor (PDFR), and its ligand (PDF) are critical for synchronizing circadian clocks and are required for robust circadian locomotor activity in *Drosophila* [139,140].

Kaneko et al. showed that the circadian clock genes *per* and *tim* are involved in regulating *Drosophila* TPR [130]. Daytime TPR is particularly prominent and exhibits robust and reproducible features controlled by clock genes. Furthermore, PER expression in DN2 neurons is necessary for daily TPR, and the PER protein is sufficient to regulate daytime TPR and does not affect locomotor activity [130]. Subsequently, Chen et al. identified two microcircuits involved in the regulation of the daytime TPR. One contains DN2 and posterior DN1 (DN1p), while the other contains anterior DN1 (DN1a). Chen et al. found that silencing DN2 neuronal activity or disrupting the clock gene *tim* in DN2 affects daytime TPR [141]. Furthermore, DN1p regulates daytime TPR independently of circadian genes. This suggests that DN1p is essential for increasing *Tp* but does not require a functional clock for TPR [141]. Live calcium brain imaging and GFP Reconstitution Across Synaptic Partners (GRASP) experiments have shown that DN2 activates DN1p to elevate daytime *Tp* and set the temperature setpoint during the daytime [141]. Additionally, DN1a is part of a DN2-independent circuit for daytime TPR, potentially integrating environmental signals such as temperature, light, and sleep information [141]. In a recent study by Alpert et al., the authors proposed that a subset of second-order thermosensory PNs, the TPN-II neurons, which are cold-processing neurons, projects to DN1a [142]. TPN-II could be upstream of DN1a; however, the exact role of TPN-II in TPR remains unclear [131].

*Drosophila Tp* is highest in the evening (ZT9-12) and decreases dramatically during the transition from daytime to night (ZT13-15) [130]. Goda et al. identified a microcircuit that controls night-onset TPR [143]. PDFR is a neuropeptide receptor that synchronizes with the circadian clock and is involved in TRP. Goda et al. found that the *Tp* of *pdfr* mutant flies increased during the daytime but decreased only by 0.8 °C at night-onset while wildtype flies exhibited a robust *Tp* decrease of 1.5–2 °C [143]. Furthermore, by rescuing *pdfr* in DN2 in a *pdfr* mutant background, the authors found that PDFR expression in DN2 neurons is sufficient for night-onset TPR [143]. Subsequently, it has been shown that PDF and diuretic hormone 31 (DH31) can bind to PDFR [144]. The rescue experiment with membrane-tethered DH31 (t-DH31) in DN2 under the DH31 mutant background demonstrates that DH31, but not PDF, regulates night-onset TPR [143]. Taken together, these studies confirmed that DH31–PDFR signaling in DN2 specifically regulates the decrease in *Tp* at night.

In the predawn phase (ZT22-ZT24), *Drosophila Tp* matches that of the early daytime (ZT1-ZT3) [130]. Tang et al. revealed that a microcircuit, AC-sLNv-DN2, is responsible for predawn TPR regulation [145]. First, by silencing neural activity and disrupting the clock gene in LNv, a PDF-positive neuron, the authors showed that LNv activity is necessary for the predawn TPR and *Tp* [145]. GRASP and live calcium brain imaging data suggest that sLNv is upstream of DN2 neurons and there are more synaptic contacts between sLNv and DN2, specifically before dawn (ZT22-24) [145]. Additionally, the neuronal functions of AC neurons as warmth sensors control the TPB and transmit temperature signals to sLNv, which activates sLNv-DN2 and sets the *Tp* before dawn [145]. In summary, these studies in *Drosophila* have uncovered the fundamental regulatory mechanisms governing TPR.

### 3.3. Feeding State-Dependent Temperature Preference

Starvation induces significant physiological changes in mammals [146,147,148,149,150]. In humans, starvation leads to a reduction in body mass, glucose levels, lipid content, and temperature [150,151,152,153]. It has been shown that hungry rats exhibit lower body mass, *Tb*, and blood sugar levels compared to sated rats [154,155,156,157]. In mice, hunger induces changes in the total cytochrome oxidase activity, GDP binding, and UCP1 protein levels, resulting in a lower *Tb* [158,159]. In *Drosophila*, there are significant *Tp* differences between sated and hungry flies. Umezaki et al. showed that flies prefer to stay in a lower-temperature environment when starved (~23 °C) compared to sated flies (~25 °C) [97].

The most well-known mechanism for transmitting satiety/hunger signals in adult flies is the *Drosophila* insulin-like peptide (Dilp). There are eight types of Dilps and one type of insulin receptor (InR) [160]. Previous studies have shown that Dilps are released from different organs in the larval or adult stages and modulate development, growth, reproduction, metabolism, stress resistance, lifespan, and cognitive functions [160]. In the adult fly brain, 14 insulin-producing cells (IPCs) are located in the pars intercerebralis (PI) and express Dilp2, Dilp3, and Dilp5 [160,161,162]. Moreover, Dilp6 is released from the fat body and suppresses Dilp2 secretion via IPCs [163]. Dilp2 and Dilp6 play key roles in delivering satiety/hunger signals in adult fly brains [163,164,165]. The Dilps–dInR interaction activates multiple signaling pathways, including the PI3K/AKT and Ras/ERK pathways, collectively known as the insulin/insulin-like growth factor signaling (IIS) pathway [166,167,168]. In the PI3K/AKT pathway, signal transduction is initiated when DILPs bind to insulin receptors (InR), triggering receptor activation through autophosphorylation. This leads to the further downstream phosphorylation of CHICO, a major substrate of InR in *Drosophila*. CHICO then binds to PI3K and induces its activation via phosphorylation at the membrane. PI3K activity is suppressed by phosphatase and tensin homolog (PTEN), which hydrolyzes PIP3 to PIP2 [166,167,168]. The accumulation of PIP3 leads to the phosphorylation and activation of protein kinases, such as *Drosophila* phosphoinositide-dependent kinase-1(dPDK1) and AKT, at the plasma membrane. AKT phosphorylates several metabolism-regulating proteins, including *Drosophila* p70 ribosomal S6 kinase (dS6K), forkhead transcription factor (FOXO), and tuberous sclerosis complex (TSC) [166,168]. In the Ras/ERK signaling pathway, DILPs bind to InR, leading to the activation of InR through autophosphorylation, which further phosphorylates downstream targets, such as Shc, initiating the MEK1/2-MAPK cascade. This cascade is typically initiated by Ras activation, which further transmits the signal by recruiting and activating MAP3K-tier Raf at the plasma membrane through phosphorylation. Raf then phosphorylates Dsor1, which, in turn, phosphorylates ERK/rolled serine/threonine kinase (MAPK). ERK is an enzyme with multiple substrates, including transcription factors and regulators of the cytoskeleton and development [169,170,171].

Two neuronal circuits modulate *Tp* in *Drosophila* under different feeding conditions. One is AC-based and the other is an MB-based neuronal circuit [65,97]. Umezaki et al. found that flies prefer low temperatures during the hunger state [97]. Using Dilps null mutants and RNAi-mediated silencing of *dilp6* in the fat body, the authors showed that Dilp6 released from the fat body is critical for starvation-induced reduction in *Tp*. It has been shown that TRPA1 in AC neurons is sufficient for controlling TPB [46]. Umezaki et al. further demonstrated that AC neurons are key regulators of starvation-induced reduction in *Tp*. Through behavioral experiments using RNAi or dominant-negative mutants of *InR*, *chico*, *PI3K*, and *PTEN*, Umezaki et al. showed that the InR signaling pathway in AC neurons plays a role in starvation-induced reduction in *Tp* [97]. Taken together, these results suggest that insulin signaling is an important mediator of starvation-induced thermoregulation via Dilp6-dInR signaling in AC neurons [97].

Chiang et al. recently demonstrated the relationship between satiety/hunger and temperature signals within MB-based neuronal circuits [65] (Figure 3B). The authors found that the feeding state significantly affects hot-avoidance behavior (HAB) in *Drosophila*. Behavioral assays revealed that hungry flies exhibit a strong HAB but not cold-avoidance behavior compared to sated flies. The authors also showed that MBn activity is critical for HAB, and α′β′ MBn is required for HAB in both sated and hungry states, while αβ and γ MBn are required for HAB only during the sated state [65]. Furthermore, the authors showed that Dilp2 is secreted from IPCs, which induces the PI3K/AKT signaling in α′β′ MBn that strongly suppresses the heat response in sated flies. Conversely, during the hungry state, Dilp2 secretion from IPCs is significantly reduced. Instead, Dilp6 is released from the fat body, which triggers the Ras/ERK signaling pathway in α′β′ MBn that shows a weak suppression of heat response in the hungry state. These results indicate that distinct insulin modulations regulate *Drosophila* HAB under different feeding conditions. Moreover, the modulated hot signals within the α′β′ MBn are further transmitted through MBON-α′3 and MBON-β′1 to execute proper HAB in both feeding states. These behavioral changes in HAB are crucial for the survival of flies. One hypothesis is that, under starvation conditions, flies conserve energy by decreasing their body temperature to reduce the rate of metabolism and become more sensitive to hot stimuli. Therefore, HAB should be increased to avoid hot environments in the hungry state. In contrast, flies become less sensitive to heat and exhibit reduced HAB, which allows them to approach higher temperatures, thereby increasing their metabolic rates after feeding [95,96].

### 3.4. Thermal Nociception Behavior

Nociception refers to the manner through which organisms react to painful or noxious stimuli that possess the capacity to inflict harm on tissues. This serves as a protective mechanism allowing animals to perceive and evade potentially detrimental stimuli in their environment. When animals encounter noxious heat that causes pain, it is known as thermal nociception. Mice exhibit licking or jumping behaviors in response to noxious heat stimuli [172]. Zebrafish larvae also exhibit increased locomotor activity when exposed to such stimuli [173,174,175]. *C. elegans* reacts with a withdrawal reflex in response to noxious temperatures [176]. Similarly, *Drosophila* larvae exhibit stereotyped escape behavior (rolling response), and adult *Drosophila* displays a rapid jump response when subjected to noxious heat stimuli [53,177,178].

Researchers have used various techniques to measure the responses of *Drosophila* larvae and adults to toxic heat. There are two primary approaches for testing how larvae react to noxious heat. An example of a well-known experiment involves placing a group of fly larvae in a Petri dish and touching them with a heated soldering iron at 46 °C. Normal larvae respond immediately by rolling [49,179]. Alternatively, larvae can be placed in a Petri dish with agar to create a film of water that allows them to move freely. The dish is then placed on a hot plate, and the rolling response of larvae is recorded at different temperatures [180]. To replicate jumping reflexes, researchers devised a method in which flies are suspended from an electric hot plate using a nylon rope. The flies are then dropped onto the plate and their jumping times are closely observed and recorded [53]. In another assay, to create the desired conditions, adult flies are placed in a specially designed incubator with bottom heating. The incubator maintains a floor temperature of 46 °C. This higher temperature discourages wild-type flies from coming into direct contact with this surface and encourages them to stay in the upper region where the temperature is around 31 °C [181]. The third method involves the integration of heat avoidance and light-seeking behavior of adult flies. This technique uses a vertical transparent apparatus equipped with a heatable metal ring and a lamp positioned at the center and the top. Flies are introduced into this apparatus. Mutant flies with diminished nociceptive behavior are attracted toward the light source and pass through the heated aluminum ring, whereas wild-type flies with intact receptors do not exhibit this behavior [182,183]. Flies respond faster to higher-intensity thermal stimuli because the delay in the jump response is inversely correlated with the thermal stimulus intensity [53].

Several pain-related genes are associated with thermal nociception. TRP channels, the largest family of noxious channels involved in pain, are best known for their functions in nociceptive perception [17]. The heat-activated channel dTRPA1 was first discovered in flies [59]. It is triggered in response to elevated temperatures, chemically reactive substances, and downstream intracellular signaling pathways [45,184,185,186]. dTRPA1 is involved in both larval and adult thermal pain [186,187]. At 46 °C, adult control flies respond relatively quickly to dangerous heat, whereas *dTrpA1* mutant flies respond more slowly and are much less able to feel heat [186]. When the temperature is below 40 °C, fly larvae begin to roll around noxiously, and the frequency of this behavior increases sharply until the temperature reaches 33 °C [187]. The dTRPA1 channel, whose activity changes in response to the rate of temperature change, is essential for all thermal nociceptive actions mentioned above.

The *painless* gene, which belongs to the TRPA channel family, plays a significant role in temperature nociception in *Drosophila* [49,53]. In larval and adult flies, *painless* is a molecular sensor for thermal stimulation [188]. Studies have shown that the *painless* mutant lacks the normal rolling behavior exhibited by wild-type larvae within one second of contact with the heated probe above 40 °C [49]. *painless* is necessary for thermal nociception in adult flies because *painless* mutant adults display behavioral defects in the hot plate assay and are unable to jump quickly to escape the noxious heat. This behavioral defect can be rescued by ectopically expressing *painless* [53]. The multidendritic sensory neurons fire more frequently when the temperature is higher than 38 °C, whereas the *painless* mutants do not show this trend [53,189]. In addition, *painless* plays a role in several neural processes in flies such as chemical and mechanical nociception [49,190], larval social behavior [191], and sexual receptivity in virgin females [51].

According to recent studies, Epi neurons respond directly to noxious heat and are necessary for reducing the response to thermal nociception. Epi neurons in the brain exhibit bilateral symmetry. These neurons extend their dendrites to innervate various regions such as the OL, LH, and areas surrounding the MBn. Additionally, the axons of Epi neurons project to segments of the ventral nerve cord (VNC), including the prothoracic, metathoracic, mesothoracic, and abdominal ganglia [192]. The percentage of jumps and the average jump latency is significantly increased and decreased, respectively, by inhibiting Epi neurons. The TRP channel pain is essential for Epi neurons to detect harmful heat levels, and mutations in TRP channels lead to an increase in thermal nociception [192]. The neuropeptide allatostatin C (AstC), released by Epi neurons, is necessary for the inhibition of thermal nociception [192,193]. AstC is similar to the mammalian neuropeptide somatostatin, which suppresses pain induced by heat [193,194]. Reducing AstC expression in Epi neurons significantly increases thermal nociception [192]. *Drosophila* has two AstC receptors: AstC-R1 and AstC-R2 [195,196]. Only AstC-R1 mutant flies exhibit increased thermal nociception [192] (Table 3).

## 4. Conclusions

*Drosophila* is an important model system for studying genetics, physiology, neurobiology, and evolutionary biology [197]. The short life span, ease of maintenance in the laboratory, and the availability of powerful genetic tools make it possible for scientists to uncover the underlying mechanisms of animal behavior, including thermosensation and thermoregulation. In this review, we comprehensively discussed how flies sense temperature, how circadian rhythms affect their temperature preferences, and how flies respond to temperatures in different internal states. We further summarized the molecules involved in *Drosophila* hot- and cold-temperature sensations (Table 1). Furthermore, we provided an overview of the primary thermosensory neurons in the peripheral nervous system that express temperature-sensitive proteins and convey temperature signals to secondary neurons (Table 2). Finally, we briefly summarized the genes and neuronal circuits involved in thermoregulation under various conditions (Table 3). Revealing the brain circuits and molecules involved in temperature-related behaviors in *Drosophila* will help in further understanding the principles of thermosensation, and pave the way for understanding the underlying mechanisms of temperature sensation in other animal species.

## Figures and Tables

**Figure 1 cells-12-02792-f001:**
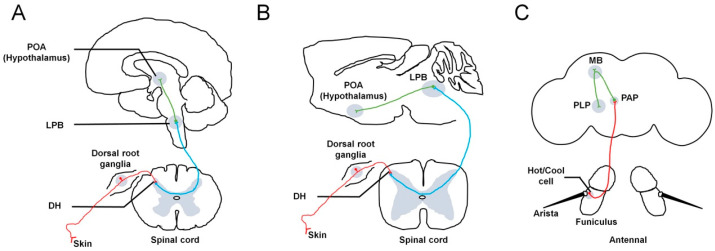
Neuronal pathways involved in thermosensation in different species. In humans (**A**) and rodents (**B**), temperature information from the skin is sensed by primary somatosensory neurons, whose cell bodies are located in the dorsal root ganglia. Subsequently, these neurons relay this information to the DH of the spinal cord (red lines). The DH contains neurons projecting to the LPB in the brainstem (blue lines). Finally, third-order neurons located within the LPB project to the POA of the hypothalamus (green line). Both warm- and cold-activated LPB neurons send projections to the POA of the hypothalamus. (**C**) In flies, the external sensors, called hot or cool cells, are located at the base of the aristae and in the sacculus region of the third antennal segment. All cool and hot cells project to a central region called the PAP to form synapses and connect with downstream PNs to convey temperature information (red line). The thermal PNs (green line) further transfer the received information to higher brain centers, including the MB and PLP. DH: dorsal horn. LPB: lateral parabrachial nucleus. POA: preoptic area. TRPP: transient receptor potential polycystin. PAP: proximal antennal protocerebrum. MB: mushroom body. PLP: posterior lateral protocerebrum. PNs: projection neurons.

**Figure 2 cells-12-02792-f002:**
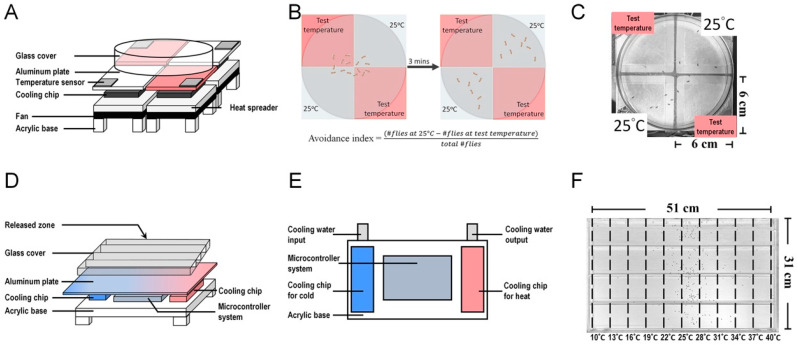
Thermoelectric devices used for temperature-related behavioral analysis in *Drosophila*. (**A**) The two-choice thermoelectric device consists of a microcontroller system, temperature sensors, aluminum plates, cooling chips, heat spreaders, and a glass cover. Pulse-width modulation (PWM) is used to control the operating voltage of the thermoelectric cooling module, which controls the temperature of the aluminum plate ranging from 15 to 35 °C. A 6 cm × 6 cm aluminum sheet is placed on each thermoelectric cooling module to increase conduction velocity. For heat dissipation, a heat spreader with an aluminum plate and a fan is placed below each cooling chip. Four thermoelectric cooling modules are arranged to form an arena, and a temperature sensor is added to each thermoelectric cooling module. (**B**,**C**) Diagram illustrating the two-choice assay behavior. Three minutes after the flies are placed in the arena, they seek a comfortable temperature environment through temperature avoidance. Flies determine their *Tp* by comparing a series of experimental temperatures with a reference temperature (25 °C) [65]. (**D**) The temperature gradient device contains an aluminum plate, which is equipped with an array of thermoelectric chips at one side for the cold source and another array of thermoelectric chips at the other side for the hot source. An electro-couple thermometer and multiple temperature sensors are set within the aluminum plate to detect and monitor the plate temperature via a microcontroller system. Water (20 °C) flows around the thermoelectric cooling chips, which supply the cold source to dissipate the accumulated heat produced by the chips. (**E**) Top view of the temperature gradient device below the aluminum plate. (**F**) *Drosophila* temperature preference behavior assay. The aluminum plate is divided into 10 sectors, each with a temperature gradient ranging from 10 to 40 °C. Thirty minutes after the flies enter the chamber, they settle at their preferred temperature locations. Flies avoid extreme temperatures and determine their *Tp* through a balance between cold- and warm-avoidance behaviors [66].

**Figure 3 cells-12-02792-f003:**
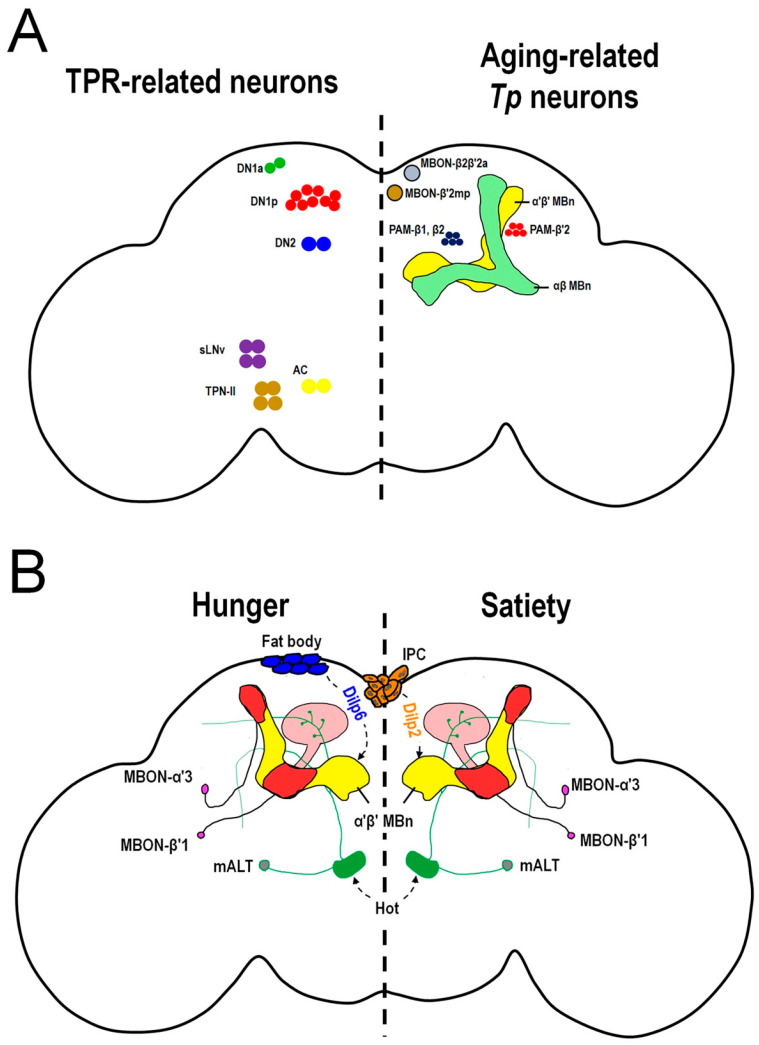
Neurons involved in temperature-related behaviors in *Drosophila*. (**A**) The TPR-related neurons (left panel) and aging-related *Tp* neurons (right panel) in the fly brain regulate *Tp.* In the left panel, there are four sets of brain microcircuits that are involved in TPR within different time zones. During daytime, two microcircuits regulate TPR, one is DN2 (blue color) > DN1p (red color) and the other is TPN-II (brown color) > DN1a (green color). During the night, DN2s (blue color) receive the DH31 ligand via PDFR to regulate TPR. In predawn, heat stimulates AC (yellow color) > sLNv (purple color) > DN2 (blue color) to regulate TPR. In the right panel, there are two brain circuits regulating aging-related *Tp*. In young flies, both the β′ system including the PAM-β′2 (red color) > α′β′ MBn (yellow color) > MBON-β′2mp (brown color) and β system including the PAM-β1, β2 (blue color) > αβ MBn (green color) >MBON-β2β′2a (gray color), contribute to normal *Tp*. However, in aged flies, the dopamine levels in PAM-β′2 are decreased which reduces the cold responses in α′β′ MBn and MBON-β′2mp. Consequently, aged flies prefer to stay at lower temperatures. The β system still functions in aged flies and is responsible for partial cold-avoidance behaviors. (**B**) The hot stimulus is conveyed through cholinergic mALT neurons (fiber: green color, soma: gray color) to α′β′ MBn dendrites and the calyx (pink color region). The α′β′ MBn (yellow color) activity is positively correlated with hot-avoidance behavior (HAB). In the food-sated state, Dilp2 is secreted from IPCs (orange color) to inhibit the hot response of α′β′ MBn by inducing the PI3K/AKT signaling pathway which causes a weak HAB. Conversely, Dilp6 is released from the fat body (blue color) to inhibit the hot response of α′β′ MBn by inducing the Ras/ERK signaling pathway. The inhibition efficiency of Ras/ERK signaling in the hungry state is not as strong as that of PI3K/AKT signaling; consequently, flies exhibit strong hot responses in α′β′ MBn during the hungry state which contributes to a strong HAB. The MBON-α′3 and MBON-β′1 (red color) play the roles of transmitting the integrated information from α′β′ MBn for proper HAB during different feeding states. TPR: temperature preference rhythm. DN1a: anterior dorsal neurons. DN1p: posterior dorsal neurons. DN2: dorsal neurons. sLNv: small ventrolateral neurons. TPN-II: second-order thermosensory projection neurons. AC: anterior cell. PAM: protocerebral anterior media. MBON: mushroom body output neuron. IPC: insulin-producing cell. Dilp: *Drosophila* insulin-like peptide.

**Table 1 cells-12-02792-t001:** Molecules involved in thermosensation in *Drosophila*.

Category	Gene	Hot/Cold	References
**TRP family**			
TRPA homologs	*dTrpA1*	Hot	[46,59,60]
	*painless*	Hot	[49]
	*pyrexia*	Hot	[54]
TRPP homologs	*brv1*, *brv2*, *brv3*	Cold	[38]
**Non TRP family**			
Rhodopsin	*rh1*	Hot/Cold	[67]
Gustatory receptor	*GR28B(D)*	Hot	[68]
Ionotropic receptor	*IR21a*	Cold	[75]
Ionotropic receptor	*IR25a*	Cold	[75]

**Table 2 cells-12-02792-t002:** Genes and circuits involved in thermosensation in *Drosophila*.

Category	Gene/Protein	Hot/Cold	Circuit	References
***Drosophila* larvae**				
DOG	IR21a/IR25a	Cold	to antennal lobes	[75,80,82,84,85]
TOG		Cold	to antennal lobes	[75,80,82,84,85]
lateral body wall		Hot/Cold		[82]
***Drosophila* adult**				
hot cell (also named warmth cell)	GR28B(D)	Hot		[68]
cool cell	*brv1*, *IR21a*, *IR25a*, *IR93a*	Cold	cool cell > fast/slow-cool-PNs	[38,86,87]
AC	*dTrpA1*	Hot		[46]
PPL1-γ1pedc		Cold	PPL1-γ1pedc > MBn	[93]
PPL1-γ2α′1		Cold	PPL1-γ2α′1 > MBn	[93]

**Table 3 cells-12-02792-t003:** Neuronal mechanisms in temperature related behaviors in *Drosophila*.

Category	Gene/Protein	Neuron/Microcircuit	References
**Aging-dependent temperature** **preference**			
early third-instar fly larvae	IR21a/IR25a/IR93a	DOCC	[105]
mid-third-instar fly larvae	*rh1*/Gq/PLC/TRPA1	*dTrpA1*-positive neuron	[67]
late-third-instar fly larvae	*rh5*/*rh6*/Gq/PLC/TRPA1; IR21a/IR25a/IR93a	*dTrpA1*-positive neuron	[104]
adult fly	Dop1R1	PAM-β1/β2 > αβ MBn > MBON-β2β′2a; PAM-β′2 > α′β′ MBn > MBON-β′2mp	[66]
**Temperature preference rhythm**			
daytime	PER	DN2s > DN1ps; TPN-II > DN1a	[130,131,141]
night-onset	PDFR/DH31	DN2	[130,131,143]
predawn	PDF/5HT1b	AC > sLNvs > DN2	[130,131,145]
**Feeding-state dependent temperature preference**			
AC-based	TRPA1/Dilp6/PI3K	fat body - AC	[97]
MB-based	Dilp2/PI3K; Dilp6/Ras	IPC/fat body - α′β′ MBn - MBON-β′2/α′3; mALT	[65]
**Thermal nociception behavior**			
fly larvae	*dTrpA1*; *painless*		[49,187]
adult fly	*dTrpA1*; *painless*	Epi neuron	[53,186,189,192]

## Data Availability

Not applicable.

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
