# Peer review of "Thermosensation and Temperature Preference: From Molecules to Neuronal Circuits in Drosophila"

_cells, 2023, doi:10.3390/cells12242792_

Round 1

Reviewer 1 Report

Comments and Suggestions for Authors

The manuscript submitted by Chiang and co-workers attempts to synthesise the available information on temperature-sensitive molecular receptors in Drosophila melanogaster. The aim is to review the molecular and neural circuits underlying thermosensation and what the authors inappropriately call "thermoregulation" (see below). Although the molecular aspects have been discussed in detail, there are several points that have not been properly reviewed or are of little relevance to the review.

- Title: The authors seem to confuse two different processes, i.e. thermoregulation and thermopreference. The former implies the active (physiological or behavioural) search for thermal equilibrium in response to a particular situation or condition. Thermopreference, on the other hand, is the expression of a preferred temperature at a given time or period. Although the authors suggest that thermopreference may be state-dependent, i.e. thermoregulatory, they do not provide a mechanistic explanation, molecular basis or neural circuits controlling thermoregulation. In the opinion of this reviewer, only the association of certain molecular receptors with thermosensation is explained, but not thermoregulation.

- Lengthy sections are devoted to thermoregulation in warm-blooded animals. This is quite irrelevant because in endotherms thermoregulation is essentially physiological and not behavioural as in ectotherms.

- Other very long sections are devoted to describing in detail experimental apparatus used by previous authors, such as thermal arenas or thermal gradients, which are well known and virtually every author has set up their own apparatus.

- In many places, the reader is left with the impression that Drosophila melanogaster is a separate entity, comparable only to vertebrates. I invite the authors to refer to the abundant literature on thermal relationships in insects, thermal adaptation, and thermoregulatory strategies and mechanisms in insects in general. In addition, fundamental concepts such as thermal performance, and critical and limiting temperatures have been largely ignored.

In summary, Drosophila melanogaster is a classic model and significant advances have been made in understanding thermoreception and neural circuits associated with thermosensation. In the opinion of this reviewer, the review should either keep the focus on these levels or perform an adequate analysis of higher level processes such as behavioural thermoregulation and thermal performances.

Author Response

Responses to Reviewer 1

The manuscript submitted by Chiang and co-workers attempts to synthesise the available information on temperature-sensitive molecular receptors in Drosophila melanogaster. The aim is to review the molecular and neural circuits underlying thermosensation and what the authors inappropriately call "thermoregulation" (see below). Although the molecular aspects have been discussed in detail, there are several points that have not been properly reviewed or are of little relevance to the review.

- Title: The authors seem to confuse two different processes, i.e. thermoregulation and thermopreference. The former implies the active (physiological or behavioural) search for thermal equilibrium in response to a particular situation or condition. Thermopreference, on the other hand, is the expression of a preferred temperature at a given time or period. Although the authors suggest that thermopreference may be state-dependent, i.e. thermoregulatory, they do not provide a mechanistic explanation, molecular basis or neural circuits controlling thermoregulation. In the opinion of this reviewer, only the association of certain molecular receptors with thermosensation is explained, but not thermoregulation.

Author’s responses:

Thank you for your comments. We have removed the discussion on “thermoregulation” from the manuscript to avoid confusion. Instead, we incorporated "temperature preference" into our title, aligning with our manuscript and behavioral descriptions. Additionally, we have provided the explanations for state-dependent thermoregulation (See Page13; Lines 564-570). We also provided comprehensive molecular basis mechanistic explanations in Page13; Lines 543-564.

- Lengthy sections are devoted to thermoregulation in warm-blooded animals. This is quite irrelevant because in endotherms thermoregulation is essentially physiological and not behavioural as in ectotherms.

Author’s responses:

In this review, we have included introductory information on thermosensation in mammals, aimed at providing general readers with a better understanding of the distinctions between warm-blooded animals and Drosophila.

- Other very long sections are devoted to describing in detail experimental apparatus used by previous authors, such as thermal arenas or thermal gradients, which are well known and virtually every author has set up their own apparatus.

Author’s responses:

Thank you for your comments. We have removed the detailed descriptions of experimental apparatus in our manuscript as suggested.

- In many places, the reader is left with the impression that Drosophila melanogaster is a separate entity, comparable only to vertebrates. I invite the authors to refer to the abundant literature on thermal relationships in insects, thermal adaptation, and thermoregulatory strategies and mechanisms in insects in general. In addition, fundamental concepts such as thermal performance, and critical and limiting temperatures have been largely ignored.

Author’s responses:

Thank you for your comments. The purpose of this review is to specifically focus on thermosensation and temperature preference behavior in Drosophila melanogaster, while excluding other insects. This review emphasizes the importance of Drosophila melanogaster as a fundamental animal model for comprehending the molecular and neuronal circuits related to thermosensation and temperature preference. Therefore, we only intended to discuss the thermosensation and temperature preference behaviors studies in Drosophila melanogaster rather than other insects.

In summary, Drosophila melanogaster is a classic model and significant advances have been made in understanding thermoreception and neural circuits associated with thermosensation. In the opinion of this reviewer, the review should either keep the focus on these levels or perform an adequate analysis of higher level processes such as behavioural thermoregulation and thermal performances.

Author’s responses:

We appreciate the valuable and constructive feedback on this manuscript. The investigation of how Drosophila engage in behavioral thermoregulation and exhibit thermal performances within higher brain centers is a significant topic, which is still under investigation. In our manuscript, we have systematically reviewed the molecular basis of how flies sense temperature (Chapter 2.1), and the neuronal basis of thermosensation (Chapter 2.2). We also addressed the mechanisms of temperature-related behaviors, such as age-related temperature preference (Chapter 3.1), temperature preference rhythm (Chapter 3.2), feeding state-dependent temperature preference (Chapter 3.3), and thermal nociception behavior (Chapter 3.4).

Reviewer 2 Report

Comments and Suggestions for Authors

In this study, the authors used Drosophila as a model to describe how temperature affects the body. The work is a wide-ranging review of the literature on issues associated with temperature related mechanisms, molecular and genetic basis of these mechanisms and, behavior presented on the example of a model organism – Drosophila melanogaster. Chapters 2. and 3. are sometimes too detailed, making the text difficult to read. Broad descriptions that do not refer directly to Drosophila unnecessarily lengthen an already very extensive text. It is worth considering whether they are necessary, it is better to remove them. The introduction lacks an explanation of the concepts of thermosensation and thermoregulation, and in the subsequent chapters there is no clear assignment of these concepts to individual parts of the manuscript. The Conclusions do not summarize the content of the chapters in a synthetic way. This part requires improvement. To sum up, from my point of view, the manuscript is valuable and qualifies for publication. Detailed comments regarding text editing are included directly in the  PDF file.

Author Response

Responses to Reviewer 2

In this study, the authors used Drosophila as a model to describe how temperature affects the body. The work is a wide-ranging review of the literature on issues associated with temperature related mechanisms, molecular and genetic basis of these mechanisms and, behavior presented on the example of a model organism – Drosophila melanogaster. Chapters 2. and 3. are sometimes too detailed, making the text difficult to read. Broad descriptions that do not refer directly to Drosophila unnecessarily lengthen an already very extensive text. It is worth considering whether they are necessary, it is better to remove them.

Author’s responses:

Thank you for your constrictive comments. In order to enhance readability, we have removed the detailed descriptions of experimental apparatus in the chapter 2 and chapter 3 in our manuscript as suggested.

The introduction lacks an explanation of the concepts of thermosensation and thermoregulation, and in the subsequent chapters there is no clear assignment of these concepts to individual parts of the manuscript. The Conclusions do not summarize the content of the chapters in a synthetic way. This part requires improvement.

Author’s responses:

Thank you for your suggestions. We have added the explanation of the concepts of thermosensation at the begging of the Introduction part as suggested. See Page 1; Lines 40-44. In the Conclusion part, we have re-mentioned all the tables, which are the summarization of the molecules and neuronal circuits of thermosensation and temperature preference in Drosophila discussed in this review article. 

To sum up, from my point of view, the manuscript is valuable and qualifies for publication. Detailed comments regarding text editing are included directly in the  PDF file.

Author’s responses:

Thank you for your helpful and constrictive comments on this manuscript.

Round 2

Reviewer 1 Report

Comments and Suggestions for Authors

The authors have adequately addressed most of my concerns and improved the manuscript accordingly. I am still unhappy about the lack of consideration of the fact that Drosophila melanogaster is an insect and should be considered as such. However, I understand that the focus of the journal is on molecular and cellular aspects.

Author Response

Reviewer 1

The authors have adequately addressed most of my concerns and improved the manuscript accordingly. I am still unhappy about the lack of consideration of the fact that Drosophila melanogaster is an insect and should be considered as such. However, I understand that the focus of the journal is on molecular and cellular aspects.

Author’s responses:

We appreciate your understanding. Indeed, we focused on discussing the molecular and cellular aspects of thermosensation in model organism, Drosophila melanogaster, in this review article. We think this fits the major goal of the special issue ” Cell Biology Research in Model Organism Drosophila”. We would like to thank you again for your helpful and constructive comments on this review article.